# Neurocognitive Inhibitory Control Ability Performance and Correlations with Biochemical Markers in Obese Women

**DOI:** 10.3390/ijerph17082726

**Published:** 2020-04-15

**Authors:** Huei-Jhen Wen, Chia-Liang Tsai

**Affiliations:** 1Center of Physical Education, College of Education and Communication, Tzu Chi University, Hualien 970301, Taiwan; 2Sports Medicine Center, Tzu Chi Hospital, Hualien 970410, Taiwan; 3Institution of Physical Education, Health and Leisure Studies, National Cheng Kung University, Tainan 701401, Taiwan

**Keywords:** obesity, inhibitory control, event-related potential, inflammatory cytokine, adiponectin/leptin ratio

## Abstract

Inhibitory control, the ability to suppress prepotent responses and resist irrelevant stimuli, is thought to play a critical role in the maintenance of obesity. However, electrophysiological performance related to different inhibitory control processes and their relationship with motor response inhibition and cognitive interference and potential biochemical mechanisms in middle-aged, obese women are as yet unclear. This work thus compared different neurocognitive Go/Nogo and Stroop task performance in healthy sedentary normal-weight and obese women, as well as their correlation with biochemical markers. Twenty-six healthy, sedentary obese women (obese group) and 26 age-matched (21–45 years old) normal-weight women (control group) were the participants, categorized by body mass index and percentage fat, as measured with dual-energy X-ray absorptiometry. They provided a fasting blood sample and performed two cognitive tasks (i.e., Go/Nogo and Stroop tasks) with concomitant electrophysiological recording. The N2 and P3 waveforms of event-related potential (ERP) were recorded. Although the between-group behavioral performance was comparable, the obese group relative to the control group showed significantly longer N2 latency and smaller P3 amplitude in the Stroop task and smaller N2 and P3 amplitudes in the Go/Nogo task. Significant inflammation response indices (e.g., CRP, leptin, adiponectin/leptin ratio) were observed in the obese group. The Nogo P3 amplitude was significantly correlated with the adiponectin/leptin ratio. These findings indicate that healthy obese women still exhibit deviant neurophysiological performance when performing Go/Nogo and Stroop tasks, where the adiponectin/leptin ratio could be one of the influencing factors for the deficit in neural processes of motor response inhibition.

## 1. Introduction

Obesity has been associated not only with non-communicable diseases, but has also been found to be related to reduced brain volume (e.g., frontal cortex and anterior cingulate cortex) and impaired neurocognitive outcomes (e.g., frontal-lobe-based executive functions) [1]. Although it is known that obesity is caused by multidimensional factors [2], evidence indicates a relationship between obesity and neural circuits related to cognitive control functions. A recent systematic review paper reported that inhibitory control is significantly impaired in obese adults and children compared to normal-weight, healthy individuals since individuals with obesity feel that they have failed to resist food (overeating behavior) [3,4].

Inhibitory control, as a subcomponent of executive functions, refers to an individual’s capacity to suppress a prepotent but ineffective behavior provoked by an external cue to stop an ongoing response and to resist distracting stimuli [5]. Such cognitive processing capabilities have been associated with many different aspects of life, being positively related to general success in health and the accumulation of wealth [6] and negatively related to addictions [7] and obesity [8]. The inhibitory control functions regulated and supported by multiple top-down neural connections [9] could be impaired in obese individuals. Indeed, obese individuals demonstrate characteristics of weak inhibitory control, which is considered to play a critical role in difficulties related to resisting external cues that suggest delicious food [10,11]. These behavioral characteristics may be associated with dopamine-modulated mesolimbic circuits and the dorsolateral regions of the prefrontal cortex [10]. In comparison with normal-weight controls, obese adults have also exhibited lower dopamine D2 receptor density in the striatum, which is associated with higher metabolic activity in the prefrontal regions involved in inhibitory control and could be a potential mechanism contributing to overeating [12].

Inhibitory control could be fractionated into two functional subcomponents: response inhibition/behavioral control (i.e., the process of countermanding a prepotent motor response) and interference control/attentional inhibition (i.e., the ability to resist interference from stimuli in the external environment) [13]. In the present study, two inhibitory tasks, Go/Nogo and Stroop, were adopted to be as homogeneous as possible in terms of visual-spatial presentation and motor requirements. The two cognitive tasks activate different task-relevant fronto-cingulo-striatal neural networks, i.e., predominantly right fronto-striatal regions during the Go/Nogo task and left-hemispheric parieto-temporal and fronto-striatal regions during the Stroop task [14]. Therefore, the neurocognitive mechanism underlying non-selective motoric stopping on the Go/Nogo task is considered to be behaviorally and neurobiologically distinct from the one implicated in response selection on the Stroop task [15]. Indeed, previous studies have used these two cognitive tasks to assess motor response inhibition (e.g., Go/Nogo task) and cognitive interference inhibition (e.g., Stroop task) in 10-year-old overweight/obese children. The results indicated divergent behavioral and cognitive electrophysiological performance [e.g., higher reaction time (RT) variability and lower P3 amplitude in the Go/Nogo task and longer RTs in the Stroop task] between normal-weight and overweight/obese groups [16].

An EEG is ideal for capturing the rapid processes involved in inhibitory control. The N2 and P3 event-related potential (ERP) components are commonly studied in relation to inhibitory control [17,18,19]. N2, an early negative deflection occurring around 200-400 ms post-stimulus, is mainly associated with conflict monitoring processes [20,21]. The following P3 wave, a positive component occurring around 300-600 ms post-stimulus, is associated with inhibition processing or attentional engagement [22,23]. The two ERP components have been demonstrated to be effective in terms of differentiating cognitive electrophysiological performance in obese and normal-weight individuals [16,24,25,26].

From the perspective of molecular biomarkers, obesity is considered to be an immunodeficient, chronic inflammation state, which may contribute to an increased risk of premature death [27]. Leptin is a pro-inflammatory cytokine produced by white adipose tissue and can cross the blood-brain barrier and blood-cerebrospinal fluid barrier via its central receptors in the hypothalamus and choroid plexus to elicit a negative feedback adiposity signal that regulates energy homeostasis in the human body [28]. Leptins have been proposed to be an early indicator of cognitive impairment [29]. Higher levels of leptin, consistent with more adipose tissue, are also associated with worse cognitive function [30,31]. Contrarily, adiponectin secreted by adipocytes has anti-inflammatory effects. Obesity and visceral adiposity are inversely correlated to adiponectin concentrations [32], normalizing following weight loss [33]. Higher adiponectin levels have been shown to be associated with cognitive performance in women [34]. In addition, the adiponectin/leptin ratio is a functional biomarker of adipose tissue inflammation [35]. Higher BMI is well-known to be associated with elevations in the peripheral levels of C-reactive protein (CRP). Poorer neuropsychological performance (e.g., slower RTs and worse inhibitory attention control) and aberrant neural activity (e.g., smaller P3 amplitudes) have been found to be associated with higher levels of CRP in adults who are overweight or obese [31].

Although inhibitory control capacity has been explored in obese children [16,36], this type of cognitive deficit should be considered from a developmental psychopathology perspective due to the protracted development of the neural networks underlying inhibitory control [37]. More importantly, although obesity is associated with a decreased ability to modulate cognitive conflict during inhibitory control processing, the neurophysiological and biochemical mechanisms underlying the relationship between inhibitory control deficits and obesity remain poorly understood. Therefore, the current study investigated the cognitive electrophysiological performance correlates of inhibitory control with motor response inhibition using the Go/Nogo task and cognitive interference inhibition using the Stroop task in sedentary obese women. In addition, since the peripheral levels of inflammatory cytokines are associated with neurocognitive performance, the correlations between biochemical markers and behavior/ERP performance when sedentary obese women performed the two cognitive tasks were also explored in the present study.

## 2. Materials and Methods

### 2.1. Ethical Approval

The participants gave informed consent prior to the study, which was approved by the Research Ethics Committee at Hualien Tzu Chi Hospital (approval number: IRB 105-61-A). This study conforms to the standards set by clause 35 of the Declaration of Helsinki, except for registration in a database.

### 2.2. Participants

With regard to Asian populations, the Western Pacific Regional Office of the World Health Organization (WPRO), led by the International Association for the Study of Obesity and the International Obesity Task Force, proposed a BMI of 23.0–24.9 kg/m^2^ for overweight and BMI ≥ 25.0 kg/m^2^ for obesity based on the related morbidity data and mortality risks [38,39]. The obese participants in the present study were thus defined as individuals with BMI ≥ 25.0 kg/m^2^. According to this criterion and after conducting an a priori power analysis, it was indicated that ~24 participants was the minimum required sample size for each group to obtain a power of 70% and moderate effect sizes [40]. Twenty-six obese adults (obese group) and 26 age-matched normal-weight (BMI < 23.0 kg/m^2^) adults (control group) aged between 21 and 45 years old were recruited from the community in Hualien City with the use of an informative flyer.

All participants were non-smokers and right handed, as measured by a handedness inventory [41]. They reported being free of any metabolic or cardiovascular diseases, neurological or psychiatric disorders, a history of head injury, or medication intake that influenced CNS functioning. None of the participants showed any symptoms of depression, as measured by the Beck depression inventory II (BDI-II; all scored below 13) [42], or cognitive impairments, as measured by the Mini-Mental State Examination (MMSE; all scored above 24) [43]. All participants had normal or corrected-to-normal vision. Demographic and body composition [e.g., BMI, circumferences, and percentage fat as measured with dual-energy X-ray absorptiometry (DXA)] data for the two groups are provided in Table 1.

### 2.3. Experimental Procedure

All participants were required to make two visits to the cognitive neurophysiology laboratory. During the first visit, the research assistant explained the experimental procedure. She then asked the participants to complete an informed consent form, as well as a medical history and demographic questionnaire, the MMSE, the BDI-II, and a handedness inventory. The height and weight of the participants were also measured to calculate their BMI. After finishing all of the questionnaires, two participants were scheduled in the morning of the same day to avoid circadian cycle bias. The participants arrived after fasting for 12 h and refraining from strenuous exercise and alcohol intake for 24 h. Each participant was asked to arrive at the laboratory at about 07.50–08.30 h and underwent fasting blood sampling by a certified technician. After blood sampling, the participant was asked to sit in an adjustable chair in front of a computer screen in an acoustically shielded room with dimmed lights and was offered two pieces of toast to avoid hypoglycaemia. An electrocap and electro-oculographic (EOG) electrodes were attached to the scalp and face of each participant before the two cognitive task tests (Go/Nogo and Stroop tasks). The viewing distance was ~75 cm. After 10 practice trials to help the participants became familiar with the protocols of the two cognitive tasks, the participants then underwent a simultaneous formal cognitive task test with concomitant electrophysiological recording, during which they were asked to respond as quickly and accurately as possible. The two cognitive tasks were performed in a counterbalanced order. All participants performed the experiment at the same time of day to control for circadian influences.

During the second visit in the same week, a DXA measurement was scheduled to assess body composition at Tzu Chi hospital. Before that, the participants were double checked for height, weight, and blood pressure.

### 2.4. Cognitive Task

#### 2.4.1. Go/Nogo Task

The Go/Nogo task [44] was presented using E-Prime 2.0 software (Psychological Software Tools, Pittsburgh, PA, USA). A “+” was shown prior to a square appearing on the screen. The participants were instructed to press the space bar as fast as possible whenever a green-colored square appeared on the screen (the Go condition) and to withhold pressing whenever a red-colored square was repeated a second time in succession (the Nogo condition). Each participant completed a practice block of 10 trials before the task began. The Go/Nogo task consisted of 200 trials (40 Nogo trials; 160 Go trials). RT was measured as the average time required to press the button after the stimuli. Accuracy rate (AR) was calculated as the percentage of correctly pressed keyboard buttons in response to both the indicative and distractor stimuli.

#### 2.4.2. Stroop Task

A two-choice version of the Stroop task programmed using E-prime (Psychology Software Tools, Sharpsburg, PA, USA), which has been demonstrated to induce a clear Stroop interference in both young and older adults [45], was adopted in this study in order to minimize the influence of response selection on task performance. In addition, given the evidence that semantics significantly interfere with color naming [46] while color interferes very little with word reading, possibly because reading is a heavily trained and highly automated process in literate adults [47,48], the color-naming condition was adopted in the present study to investigate the effect of obesity on neurocognitive functioning associated with Stroop interference. The stimuli were two color names in Chinese presented as “紅” (red) and “綠” (green). All stimuli were presented with 4.5 × 4.5 cm letters in the center of a 21-in. cathode-ray tube display against a black background at an 80 cm distance. In the incongruent condition, the color of the word in the display was different from its word meaning, whereas in the congruent condition, the meaning of the word and its color matched. A single test block consisted of 50% incongruent and 50% congruent trials, in a randomized order. Two blocks of 120 trials, for a total 240 trials, were presented to each participant, with a rest period of 2 min between blocks. Each stimulus appeared on the screen until the participant responded and the next stimulus appeared 1.5 to 2 s after the response. The participants were instructed to respond as quickly and accurately as possible with a button press from their index (“N” key) and middle (“M” key) fingers of their right hand on a computer in response to the color while ignoring the word meaning. The stimulus response pairs were counterbalanced across participants. All participants performed the Stroop task with simultaneous electrophysiological recording. After a practice block of 10 trials to ensure that the participants understood the task instruction, the formal test was administered to allow for collection of behavioral performance along with EEG data.

### 2.5. Whole and Regional Body Composition

Body composition was measured using dual energy X-ray absorptiometry (DXA; Discovery Wi, Hologic Inc., Bedford, MA, USA). The measurement was performed by a certified technician according to the standard operating procedure. The scanning instructions and procedures were standardized for all participants. The trunk region included the area from the bottom of the neckline to the top of the pelvis, excluding the arms. The mass output from the DXA scanner was expressed in grams. The accuracy of the densitometer was calibrated using the manufacturer’s spine phantom with a known hydroxyapatite density each testing day.

### 2.6. ERP Recording and Analysis

The ERP was recorded using eego^TM^ amplifier system (ANT Neuro, EE-211, revision Nr 1.2, Germany) from 64 scalp sites (10–10 system) with Ag/AgCl electrodes (active electrodes) mounted in an elastic cap. All inter-electrode impedance was maintained below 5 KΩ. The raw EEG signal was acquired with an A/D rate of 500 Hz/channel, a band-pass filter of 0.1–50 Hz, and a 60-Hz notch filter. An offline electrooculographic correction was applied to the individual trials prior to averaging the ERP components. All trials with response error and artefacts (i.e., electrooculogram and electromyogram exceeding ± 100 µV) were discarded. The remaining effective ERP data were separately averaged offline and constructed from Go and Nogo conditions in the Go/Nogo task and from congruent and incongruent conditions in the Stroop task over a 1000 ms epoch beginning 200 ms prior to the onset of the target stimulus. The mean amplitudes and latencies of the N2 and P3 components were measured at the Fz, FCz, and Cz electrodes. The time windows for detection of the N2 and P3 components were 150–350 ms and 350–600 ms, respectively. Latency was calculated as the time in milliseconds from stimulus onset to peak amplitude.

### 2.7. Blood Sampling and Analysis

Fasting blood samples were collected to determine the levels of CRP, leptin, and adiponectin. The participants were instructed to fast overnight and abstain from caffeine and alcohol for 12 h prior to blood sampling. A blood sample of approximately 10 mL was collected into an EDTA vacutainer tube via venipuncture in the antecubital fossa. The blood samples were centrifuged at 1000 g for 8 min at 4 °C within 30 min of collection. The plasma fraction was aliquoted in storage tubes and stored at −80 °C until analysis. The inflammatory biomarkers were assayed using commercially available kits according to the manufacturer’s protocols. Plasma CRP (REF 378020, Beckman coulter Inc., CA, USA) was determined using a commercial enzyme-linked immunosorbent assay (ELISA) kit (UniCel^®^ DxC 600/800 System(s) and SYNCHRON^®^ Systems CAL 5 Plus, Beckman Coulter, Inc.). Plasma leptin (250 tubes, Cat. # HL-81K, the Linco Research, Inc., Billerica, MO, USA) and adiponectin levels (125 tubes, Cat. # HADP-61HK, the Linco Research, Inc., Billerica, MO, USA) were measured using a standardized RIA kit for humans (LINCO Research Inc., Chaska, MN, USA). All samples were assayed in duplicate and the mean of the two duplicate values was used in the statistical analyses.

### 2.8. Data Processing and Statistical Analyses

Repeated measure analyses of variance (RM-ANOVAs) were conducted using group (obese vs. control) as a between-subject factor. For the behavioral [RTs and accuracy rates (ARs)] data, the conditions were conducted as a within-subject factor (congruent vs. incongruent for the Stroop task; Go vs. Nogo for the Go/Nogo task except the RTs). For the ERP data, the electrode (Fz vs. FCz vs. Cz) was also examined as a within-subject factor. Bonferroni post-hoc analyses were performed when there were significant differences. Partial Eta squared (*η_p_*^2^) was adopted to calculate effect sizes for significant main effects and interactions, with the following criteria used to determine the magnitude: 0.01–0.059 indicated a small effect size; 0.06–0.139 indicated a medium effect size; and > 0.14 indicated a large effect size. Correlations were used to analyze the relationships between the neurophysiological indices and the biomechanical biomarkers. Statistical analyses were conducted with SPSS software version 24.0 (SPSS Inc., Chicago, IL, USA). A *p*-value ≤ 0.05 was considered statistically significant.

## 3. Results

### 3.1. Demographic Data

As shown in Table 1, the weight and circumference measures (e.g., BMI, waist girth, abdominal girth, and hip girth) differed significantly between the obese and control groups. Significant between-group differences in the body composition status were only found for whole body fat and upper limb fat percentages. The blood pressure (e.g., SBP and DBP) and bone health (e.g., BMD and T-score) fell in the normal range even though the data showed significant between-group differences. In addition, no significant differences were found in the values for the other demographic measures.

### 3.2. Behavioral Performance

The behavioral performance of the normal-weight and obese women is shown in Table 2. The numbers of usable trials are reported in the Appendix A. Obese and control groups did not show a significant difference in the average number of trials for RTs (Go/Nogo task: Go trials, *p* = 0.751 & Nogo trials, *p* = 0.790; Stroop task: congruent trials, *p* = 0.473 & incongruent trials, *p* = 0.502).

#### 3.2.1. Go/Nogo Task

Accuracy rate (AR)

The RM-ANOVA for the ARs showed a significant main effect of *condition* [F_(1,50)_ = 19.07, *p* < 0.001, *η_p_*^2^ = 0.29], with the AR being significantly higher in the Go condition (99.91 ± 0.22%) than in the Nogo condition (98.27 ± 2.61%). No significant main effects of *group* [F_(1,50)_ = 0.02, *p* = 0.887] or significant interaction between *group* and *condition* [F_(1,50)_ = 0.04, *p* = 0.835] were found.
Reaction time (RT)

A univariate variation analysis revealed no significant differences in the RT for *group* [F_(1,50)_ = 0.69, *p* = 0.411] in the Go condition.

#### 3.2.2. Stroop Task

Accuracy Rate (AR)

The RM-ANOVA for the ARs showed a significant main effect of *condition* [F_(1,50)_ = 27.76, *p* < 0.001, *η_p_*^2^ = 0.36], with the AR being significantly higher in the congruent condition (97.13 ± 13.80%) than in the incongruent condition (95.42 ± 13.79%). No significance main effects of *group* [F_(1,50)_ = 1.02, *p* = 0.306] or significant interactions between *group* and *condition* [F_(1,50)_ = 0.79, *p* = 0.379] were found.
Reaction time (RT)

The RM-ANOVA for the RTs showed a significant main effect of *condition* [F_(1,50)_ = 60.83, *p* < 0.001, *η_p_*^2^ = 0.55], with the RT being significantly faster in the congruent condition (523.50 ± 59.90 ms) than in the incongruent condition (568.35 ± 90.09 ms). No significant main effects of *group* [F_(1,50)_ = 0.01, *p* = 0.925] or significant interactions between *group* and *condition* [F_(1,50)_ = 0.19, *p* = 0.663] were found. The value obtained when subtracting inhibitory control in the congruent RTs from the incongruent RTs showed a non-significant between-group difference [control vs. obese: 42.37 vs. 47.42 ms, t = −0.44, *p* = 0.663].

### 3.3. Electrophysiological Performance

The numbers of usable trials are reported in the Appendix A. Obese and control groups did not show a significant difference in the average number of trials for ERP analysis (Go/Nogo task: Go trials, *p* = 0.542 and Nogo trials, *p* = 0.654; Stroop task: congruent trials, *p* = 0.908 and incongruent trials, *p* = 0.935).

#### 3.3.1. Go/Nogo Task

Figure 1 and Figure 2 display the grand-average ERP waveforms for the three midline electrodes during the Go/Nogo task and the Stroop task in the two groups.
N2 component

As shown in Figure 1, the RM-ANOVA for the N2 latency revealed a significant main effect of ***condition*** [F_(1,50)_= 4.65, *p* = 0.036, *η_p_*^2^ = 0.09], with the Go condition (309.30 ± 58.81 ms) having a shorter N2 latency than the Nogo condition (327.08 ± 40.62 ms). No significant main effects of *group* [F_(1,50)_= 1.18, *p* = 0.284] and *electrode* [F_(2,104)_= 1.87, *p* = 0.178] or significant interactions among *group*, *condition*, and *electrode* were observed [*group × condition*: F_(1,50)_= 0.14, *p* = 0.713; *group* × *electrode*: F_(2,100)_ = 0.31, *p* = 0.580; *group* × *condition* × *electrode*: F_(2,98)_ = 0.35, *p* = 0.558].

The RM-ANOVA for the N2 amplitude showed that there were significant main effects of *group* [F_(1,50)_ = 5.52, *p* = 0.023, *η_p_*^2^ = 0.10] and *condition* [F_(1,50)_ = 20.33, *p* < 0.001, *η_p_*^2^ = 0.29], with the control group (−3.34 ± 2.18 μV) exhibiting a smaller N2 amplitude than the obese group (−2.12 ± 1.43 μV, *p* = 0.023) and the Go condition (−2.30 ± 1.88 μV) showing a greater N2 amplitude than the Nogo condition (-3.18 ± 2.22 μV, *p* < 0.001) across the two groups. No significant main effects of *electrode* [F_(2,104)_ = 0.01, *p* = 0.914] or significant interactions among *group*, *condition*, and *electrode* were observed [*group* × *condition*: F_(1,50)_= 0.00, *p* = 0.964; *group* × *electrode*: F_(2,100)_ = 0.88, *p* = 0.352; *group* × *condition* × *electrode*: F_(2,98)_ = 1.22, *p* = 0.275].
P3 component

The RM-ANOVA for the P3 latency revealed that there were no main effects of *group* [F_(1,50)_ = 1.83, *p* = 0.182], *electrode* [F_(2,104)_ = 1.33, *p* = 0.270], and *condition* [F_(1,50)_= 0.97, *p* = 0.329]. No significant interactions among *group*, *condition*, and *electrode* were found [*group* × *condition*: F_(1,50)_ = 1.49, *p* = 0.228; *group* × *electrode*: F_(2,100)_ = 0.01, *p* = 0.937; *group* × *condition* × *electrode*: F_(2,98)_ = 0.41, *p* = 0.526].

The RM-ANOVA for the P3 amplitude revealed significant main effects of *group* [F_(1,50)_ = 19.19, *p* < 0.001, *η_p_*^2^ = 0.28], *condition* [F_(1,50)_ = 667.41, *p* < 0.001, *η_p_*^2^ = 0.93], and *electrode* [F_(2,104)_ = 56.75, *p* < 0.001, *η_p_*^2^ = 0.54], with the control group (2.98 ± 0.53 μV) exhibiting greater P3 amplitudes than the obese group (2.34 ± 0.51 μV), the Go condition (1.11 ± 0.30 μV) showing greater P3 amplitudes than the Nogo condition (4.22 ± 1.03 μV) across the two groups, and the Cz site (2.93 ± 0.69 μV) showing greater P3 amplitude than that of the Fz site (2.19 ± 0.77 μV). There were also significant main effects for *group* × *condition* [F_(1,50)_= 5.79, *p* = 0.020, *η_p_*^2^ = 0.11] and *group* × *electrode* interactions [F_(2,100)_ = 13.37, *p* = 0.001, *η_p_*^2^ = 0.21], with the obese group showing smaller P3 amplitudes than the control group in the *Go* (0.93 ± 0.26 μV vs. 1.28 ± 0.22 μV, *p* < 0.001) and *Nogo* (3.74 ± 0.93 μV vs. 4.67 ± 0.92 μV, *p* = 0.001) conditions, with the obese relative to the control group exhibiting smaller P3 amplitudes at the Fz (1.67 ± 0.65 μV vs. 2.69 ± 0.49 μV, *p* < 0.001) and FCz (2.57 ± 0.60 μV vs. 3.16 ± 0.68 μV, *p* = 0.001) sites. However, no significant interactions among *group*, *condition*, and *electrode* were observed [*group* × *condition* × *electrode*: F_(2,98)_ = 1.86, *p* = 0.179].

Since blood pressure is a confounding factor with regard to cognitive electrophysiological performance of visuospatial attention in adults [49] and the obese group had significantly higher SBP and DBP compared to the normal-weight control group (see Table 1), the ERP components, N2 and P3 amplitudes, were used to account for blood pressure using an analysis of covariance (ANCOVA) procedure. The results of the RM-ANCOVA on the P3 amplitude across the two conditions still indicated a significant main effect of *group* [F_(1,48)_ = 14.85, *p* < 0.001, *η_p_*^2^ = 0.24]. However, the results of RM-ANCOVA on the N2 amplitude across the two conditions indicated a non-significant main effect of *group* [F_(1,48)_ = 3.83, *p* = 0.056].

#### 3.3.2. Stroop Task

N2 component

As shown in Figure 2, the RM-ANOVA for the N2 latency showed a significant main effect of *group* [F_(1,50)_= 5.09, *p* = 0.029, *η_p_*^2^ = 0.098], with the obese group (358.49 ± 9.07 ms) exhibiting a longer N2 latency than the control group (274.85 ± 9.46 ms). No significant main effects of *condition* [F_(1,50)_ = 0.22, *p* = 0.644] and *electrode* [F_(1,50)_= 0.05, *p* = 0.829] or significant interactions among *group*, *condition*, and *electrode* were observed [*group* × *condition*: F_(1,50)_= 0.25, *p* = 0.622; *group* × *electrode*: F_(2,100)_ = 2.97, *p* = 0.092; *group* × *condition* × *electrode*: F_(2,98)_ = 1.36, *p* = 0.249].

In terms of N2 amplitude, there were no significant main effects for *group* [F_(1,50)_ = 0.67, *p* = 0.417], *electrode* [F_(2,104)_ = 0.01, *p* = 0.914], and *condition* [F_(1,50)_ = 0.13, *p* = 0.720]. No significant interactions among *group*, *condition*, and *electrode* were observed [*group* × *condition*: F_(1,50)_= 1.33, *p* = 0.255; *group* × *electrode*: F_(2,100)_ = 0.01, *p* = 0.944; *group* × *condition* × *electrode*: F_(2,98)_ = 0.03, *p* = 0.858].P3 component

The RM-ANOVA for the P3 latency showed that there was a significant main effect of *group* [F_(1,50)_ = 6.14, *p* = 0.017, *η_p_*^2^ = 0.12], with the obese group (405.26 ± 60.84 ms) exhibiting a longer P3 latency than the control group (368.29 ± 42.99 ms). No significant main effects of *electrode* [F_(2,104)_ = 0.76, *p* = 0.473] and *condition* [F_(1,50)_ = 0.16, *p* = 0.689] were observed. No significant interactions among *group*, *condition*, and *electrode* were observed [*group* × *condition*: F_(1,50)_ = 0.35, *p* = 0.555; *group* × *electrode*: F_(2,100)_ = 0.84, *p* = 0.365; *group* × *condition* × *electrode*: F_(2,98)_ = 0.04, *p* = 0.852].

In terms of the P3 amplitude, there were significant main effects of *group* [F_(1,50)_ = 1514.68, *p* <.001, *η_p_*^2^ = 0.97], *condition* [F_(1,50)_ = 317.02, *p* < 0.001, *η_p_*^2^ = 0.871], and *electrode* [F_(2,104)_= 1097.80, *p* < 0.001, *η_p_*^2^ = 0.96], with the control group (3.11 ± 0.18 μV) exhibiting a greater P3 amplitude than the obese group (1.64 ± 0.05 μV) and greater P3 amplitude in the congruent condition (2.72 ± 0.82 μV) than in the incongruent condition (2.12 ± 0.70 μV) across the two groups, with the P3 amplitude at the Cz (3.34 ± 0.72 μV) > FCz (2.40 ± 0.94 μV) > Fz (1.51 ± 0.65 μV). There were significant interactions among *group*, *condition*, and *electrode* [*group* × *condition*: F_(1,50)_= 7.31, *p* = 0.010, *η_p_*^2^ = 0.14; *group* × *condition* × *electrode*: F_(2,98)_ = 6.73, *p* = 0.013, *η_p_*^2^ = 0.13]. The post hoc analysis showed that the P3 amplitudes at all electrodes (Fz: 1.16 ± 0.07 μV; FCz: 1.51 ± 0.02 μV; Cz: 2.98 ± 0.06 μV) in the congruent condition in the obese group were significantly smaller than in the control group (Fz: 2.33 ± 0.17 μV; FCz: 3.64 ± 0.16 μV; Cz: 4.39 ± 0.83 μV). Identically, the P3 amplitudes at all electrodes (Fz: 0.51 ± 0.01 μV; FCz: 1.31 ± 0.02 μV; Cz: 2.33 ± 0.21 μV) in the incongruent condition in the obese group were significantly smaller than in the control group (Fz: 1.89 ± 0.10 μV; FCz: 2.91 ± 0.20 μV; Cz: 3.49 ± 0.16 μV).

Since blood pressure is a confounding factor with regard to cognitive electrophysiological performance of visuospatial attention in adults [49] and the obese group had significantly higher SBP and DBP compared to the normal-weight control group (see Table 1), the ERP components, N2 and P3 amplitudes, were used to account for blood pressure using an analysis of covariance (ANCOVA) procedure. The results of the RM-ANCOVA on the N2 and P3 amplitudes across the two conditions still indicated a significant main effect of *group* [N2: F_(1,48)_ = 831.56, *p* < 0.001, *η_p_*^2^ = 0.95; P3: F_(1,48)_ = 7.44, *p* = 0.010, *η_p_*^2^ = 0.14].

### 3.4. Biochemical Indices

As shown in Figure 3, the levels of the biochemical indices (i.e., CRP, leptin, adiponectin, and adiponectin/leptin ratio) exhibited substantial *group* differences (*ps* < 0.05 in all cases). The obese group exhibited higher levels in inflammatory biomarkers (CRP and leptin) as compared to the control group (*p*s < 0.05). In addition, the levels of adiponectin and the adiponectin/leptin ratios were significantly lower in the obese group than in the control group.

### 3.5. Correlations

#### 3.5.1. Go/Nogo Task

In the obese group, only the P3 amplitude across the three electrodes in the Nogo condition was significantly correlated with the adiponectin/leptin ratio (r = 0.40, *p* = 0.049).

#### 3.5.2. Stroop Task

Correlations between neurocognitive performance and the biochemical markers (i.e., CRP, leptin, adiponectin, and adiponectin/leptin ratio) in the obese group were not found.

## 4. Discussion

The aim of the present study was to explore the neurophysiological and molecular biochemical mechanism of executive control problems in obese women when performing two cognitive tasks involving different inhibitory control processes and to investigate the relationships between neurocognitive performance and the biochemical markers under consideration. The main findings indicated that although the obese and normal-weight women showed comparable ARs and RTs in the Stroop and Go/Nogo tasks, they exhibited longer N2 and P3 latencies and smaller P3 amplitudes across both conditions in the Stroop task and larger N2 amplitudes across both conditions and smaller P3 amplitudes in both the Go and Nogo conditions in the Go/Nogo task. Only the Nogo P3 amplitude in the obese group was positively correlated with the adiponectin/leptin ratio.

### 4.1. Neurocognitive Performance

In the present study, obese women showed comparable ARs and RTs to those of the normal-weight women in both the Stroop and Go/Nogo tasks, suggesting that obesity did not impact behavioral performance when they performed the two cognitive tasks with different inhibitory control processes. The present findings were somewhat inconsistent with earlier studies in which overweight/obese children relative to their normal-weight counterparts exhibited similar ARs, slower RTs in the Stroop task, and higher RT variability in the Go/Nogo task, suggesting impaired inhibitory control functions and a reduction in response efficiency in obese children [16]. In addition, Tsai et al. (2016) also found that obese children showed significantly weaker inhibitory control of attention when performing a visuospatial attention task [36]. The lack of age-related effects on behavioral performance in these inhibitory-control-related tasks in the present and previous studies may be related to core cognitive processes that are in place early in development [50]. Although children and adolescents appear fully mature in their ability to recognize when they have made an error, activity in the dorsal anterior cingulate cortex, which is associated with correct trial performance, in these age groups differed from that in adults [50,51]. The development of inhibitory control as indicated from the brain activity of the caudate body using Go/Nogo and Stroop tasks increases from childhood to adulthood [50,52]. Hence, the performance of children and adolescents receives less support from such feedback signaling, implicating immature error regulation and error feedback utilization as a source of performance decrements at younger ages [50]. This could be a possible explanation for the contrasting behavioral results between children and adults related to obesity. Since the emergence of adult-level cognition is known to rely on the development of error-regulatory functions, particularly in the context of being able to voluntarily inhibit responses to task-irrelevant stimuli, the present study extended previous knowledge by showing that obese women exhibited consistent ARs and RTs, suggesting that their cognitive control is more stable and less prone to external influences [53].

The Stroop task, which predominantly addresses the left-hemispheric fronto-striatal and parieto-temporal regions [14], assesses cognitive interference inhibition, while the Go/Nogo task, by activating task-relevant fronto-cingulo-striatal neural networks, evaluates motor response inhibition [14]. This study showed that obese women relative to normal-weight women exhibit longer Stroop N2 and P3 latencies in the Stroop task, but not in the Go/Nogo task. These results are compatible with previous findings of a significant discrepancy in P3 latency in the Stroop task in adolescents [24] and a non-significant discrepancy between N2 and P3 latencies in the Go/Nogo task between obese children/adolescents and control groups [16,54]. Therefore, both previous studies and the present study presumably reflect dysfunction within the anterior cingulate or prefrontal cortex (i.e., longer N2 and P3 latencies in the Stroop task) [55,56,57] in cases of obesity, while task-relevant fronto-cingulo-striatal neural networks seem to be somewhat intact (i.e., comparable N2 and P3 latencies between obese and control groups in the Go/Nogo task).

Obese women produced a greater Go/Nogo N2 amplitude than normal-weight women in the current study, which concurred with earlier studies on obese adolescents [54], supporting that individuals suffering from obesity show neural deficits in underlying systems modulating conflict monitoring processes [20,25,54,58]. However, the deviant N2 amplitude between the two groups only approached significant when controlling for the blood pressure as a co-variate, suggesting that blood pressure could be a potential factor related to inhibitory control deficits in individuals with obesity when performing the Go/Nogo task. Nevertheless, obese individuals have been demonstrated to show compromised inhibitory control processes that are involved in multiple top-down neural connections [9,16]. Therefore, the prefrontal cortex can be viewed as a predictor, a mediator, or a causal agent of obesity [59].

P3 amplitude can be regarded as an index of a motivation-related attentional engagement [26,60]. Although Nijs et al. (2010) and Bauer et al. (2010) reported no significant differences between obese and normal-weight individuals for P3 amplitudes in different Stroop tasks [24,26], most studies have found that obese individuals exhibit reduced P3 amplitudes in the Go/Nogo task [16,25,61]. In the present study, smaller P3 amplitudes were observed not only in the Go/Nogo task, but also in the Stroop task in the obese women as compared to the normal-weight women. These findings support the premise that obese individuals compromise a later stage of the inhibitory process when actual inhibition of the motor system in the premotor cortex takes place [62,63]. Since P3 amplitude also represents more conscious and controlled attentional processing of information, the deviant P3 neural activity occurring in obese individuals could reflect fewer attentional resources being allocated to inhibitory control processes and greater difficulty in processing stimuli, which results in a problem related to efficiently recruiting the cognitive resources necessary to modulate the risk of overeating behavior followed by weight gain [64].

### 4.2. Molecular Biomarkers and Correlations with Neurocognitive Performance

In the present study, as expected, obese women exhibited higher levels of CRP and leptin, lower levels of adiponectin, and lower adiponectin/leptin ratios as compared to the normal-weight women. Our results were consistent with previous findings [30,31]. Obesity alone, in the absence of overt disease, is frequently accompanied by subclinical systemic inflammation marked by increased circulating levels of proinflammatory indicators such as CRP and leptin [65,66]. These two inflammatory proteins play a key role not only in regulating food intake and energy expenditure and enhancing insulin sensitivity [30,67], but also in influencing hypothalamic function [30,68] and neuronal degradation [69] and decreasing neuronal excitability [70]. The associations between leptin/CRP levels and neurophysiological performance in the current study were not observed in the obese women. These findings differ from those of a previous study in which ERP P3 amplitude was negatively correlated with both the levels of leptin and CRP in obese individuals when performing a visuospatial attention task [31], but partly concur with studies in which there were significant associations between leptin levels and neuropsychological performance in individuals with mild cognitive impairment. Obesity is well recognized as a state of inflammation in which increased levels of inflammatory biomarkers may interfere with leptin receptors and inhibit the neuroprotective effect on the brain. Higher levels of leptin have not been found to confer any neuroprotective benefit for overweight or obese women [71]. Additionally, from the point of view of gender differences, previous studies reported that leptin levels are not correlated with cognitive performance in women [72,73], but are in men [74].

The Nogo P3 amplitude in the obese women in the present study was positively correlated with the adiponectin/leptin ratio. To date, no previous studies have explored the relationship between the adiponectin/leptin ratio and neurocognitive performance in obese individuals. There is a possible explanation to account for the pattern of the findings on this topic. Adiponectin acts as an insulin-sensitizing hormone in muscle and liver, and lower levels of adiponectin further contribute to peripheral insulin resistance in obesity [75,76]. The adiponectin/leptin ratio is a functional biomarker of adipose tissue inflammation [77] and a good indicator of a dysfunctional adipose tissue, which may be a useful estimator of obesity and metabolic syndrome [77]. This emerging biomarker correlates with insulin resistance better than adiponectin or leptin alone [77], while insulin resistance correlates with cognitive dysfunction [78,79,80]. Since individuals with type 2 diabetes suffer from severe cognitive deficits [79] and deficits in hippocampal function may appear due to peripheral insulin resistance and hyperlipidemia caused by a high-calorie diet, which declines hippocampal synaptic plasticity and impairs cognitive function [80], this is a plausible reason for the significant correlation between the adiponectin/leptin ratio and the neurophysiological performance observed in the present study.

### 4.3. Limitations

Although this is the first study to examine the neurocognitive inhibitory control ability performance and correlations with biochemical markers in obese women, where the confounding factors were rigorously controlled, there are still some potential limitations to the cross-sectional study design. First, an earlier study reported that there might be an effect of obesity on cognitive function that is only observable in men [81]. The association between adiposity-related indices (e.g., BMI and body fat %) and low-grade systemic inflammation (e.g., CRP and leptin) is considerably stronger in women than in men [82,83,84]. Also, visceral fat cannot completely explain the levels of adiponectin in women as compared to men [85]. Therefore, gender plays a significant role in pathophysiological changes and clinical manifestations due to a crucial effect of sex hormones on neurohumoral adipose tissue activity [86]. Future studies could examine these gender differences to re-evaluate neurocognitive performance and biochemical findings related to obesity. Second, thus far, the consensus has not been reached concerning the relationships between BMI/adiposity and cerebral cortical thickness [87,88,89]. An avenue for future work is to examine the possibility of whether the ERP signal could be affected by the possible adipose tissue differences on the scalp between obese and healthy-weight individuals. Third, ERP N2 and P3 components are affiliated with cognitive processes controlled at the levels of the anterior cingulate cortex and hippocampal and parietal cortical region [90,91]. Previous studies demonstrated that physical lesions (e.g., brain cortex and tissue) could affect the ERP waveform [91,92]. Therefore, deviant neurophysiological performance in the obese individuals in the present study could be attributed to brain structure being affected by obesity, not worse cognitive processes. This conjecture is somewhat speculative, but provides a basis for future research.

## 5. Conclusions

Healthy obese women relative to normal-weight women showed comparable behavioral indices when they performed the Go/Nogo and Stroop tasks in the present study. However, they still exhibited deviant neurophysiological performance in underlying neural systems modulating motor response inhibition and cognitive interference inhibition and higher levels of inflammatory cytokines. Regular exercise has been demonstrated to be effective to improve higher levels of inflammatory cytokines and compromised neural activity in obese adults [93]. Determining how to control the adiponectin/leptin ratio through a healthy lifestyle (e.g., performing physical activity and exercise) to mitigate the deficit in neural processes of motor response inhibition could be an important issue in clinical practice.

## Figures and Tables

**Figure 1 ijerph-17-02726-f001:**
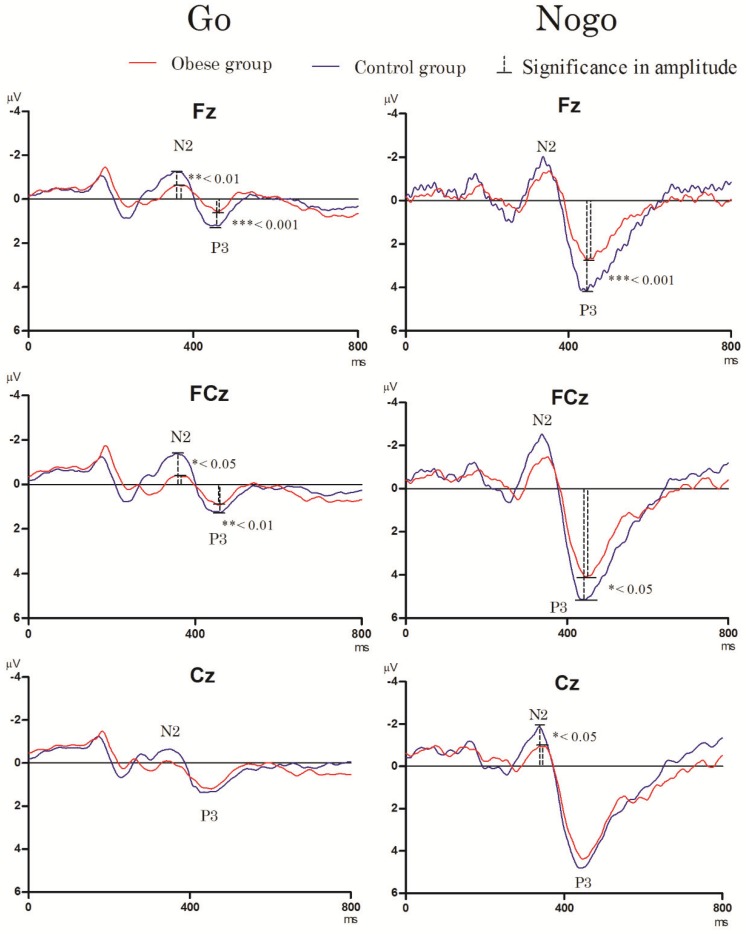
Grand averaged ERPs of N2 and P3 waveforms in the Go and Nogo conditions in three electrodes (Fz, FCz, and Cz) for the normal-weight control group (CG) and the obese group (OG) in the Go/Nogo task.

**Figure 2 ijerph-17-02726-f002:**
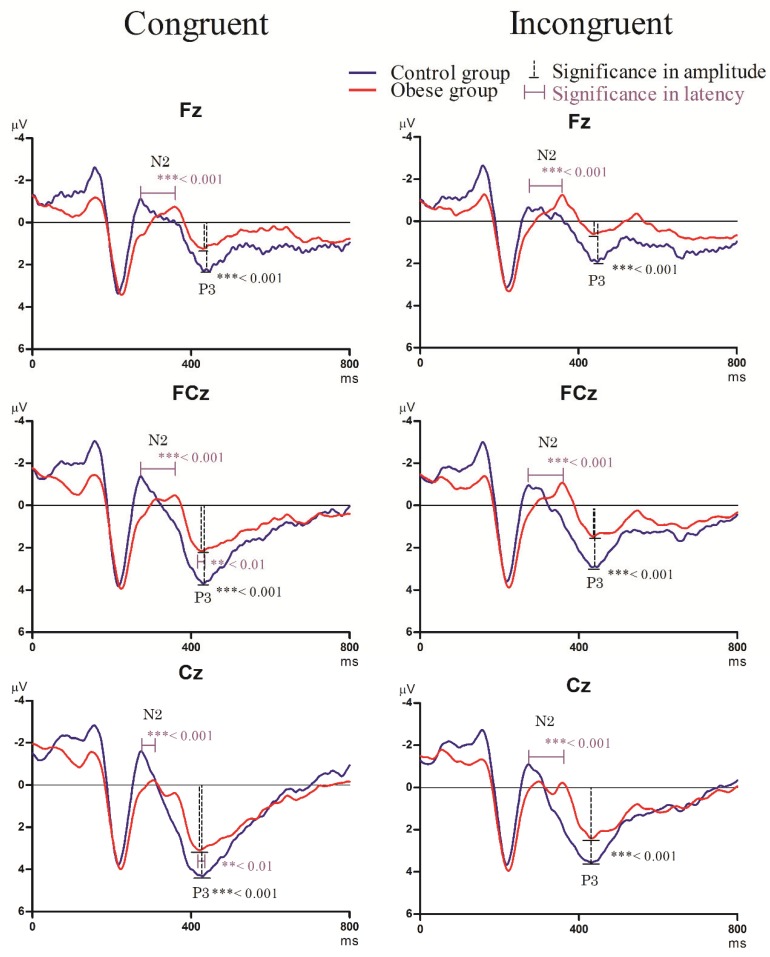
Grand averaged ERPs of N2 and P3 waveforms in the congruent and incongruent conditions in three electrodes (Fz, FCz, and Cz) for the normal-weight control group (CG) and the obese group (OG) in the Stroop task.

**Figure 3 ijerph-17-02726-f003:**
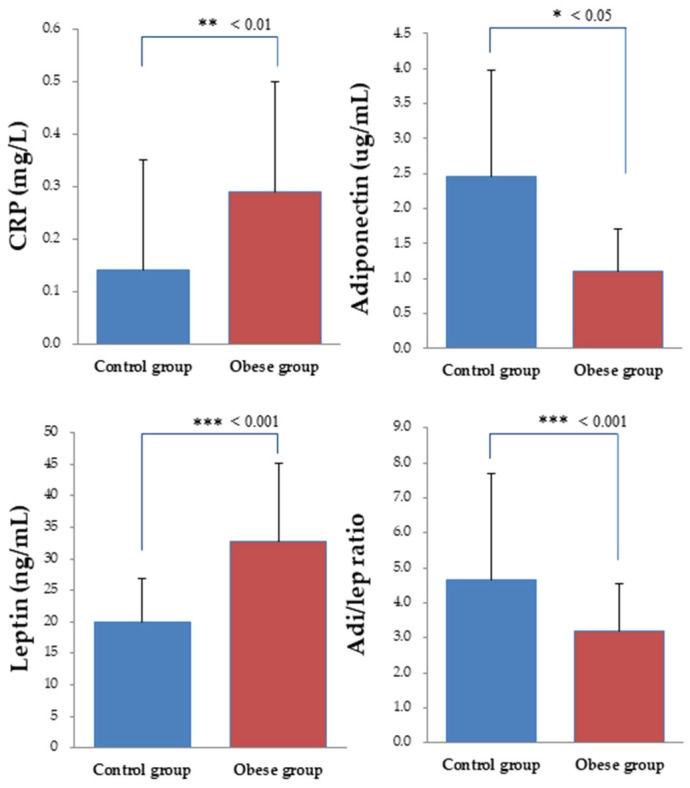
Biochemical values of the normal-weight control group and the obese group.

**Table 1 ijerph-17-02726-t001:** Demographic characteristics of the normal-weight (control) and obese women.

Characteristics	Control Group (n = 26)	Obese Group (n = 26)	*t*	*p* Value
Age (years)	34.04 ± 5.66	34.44 ± 5.77	−0.25	0.802
Height (cm)	161.06 ± 6.31	160.32 ± 5.39	0.45	0.653
Weight (kg) *	58.89 ± 4.49	75.80 ± 11.43	−7.00	<0.001
BMI (kg/m^2^) *	22.70 ± 1.13	29.41 ± 3.52	−9.24	<0.001
SBP (mmHg) *	106.46 ± 14.26	115.60 ± 12.46	−2.43	0.019
DBP (mmHg) *	68.00 ± 9.43	76.56 ± 9.11	−3.30	0.002
Resting HR (bpm)	72.23 ± 7.64	75.52 ± 7.01	−1.60	0.116
Education (years)	16.38 ± 0.80	16.00 ± 1.41	1.20	0.236
BDI-II	11.08 ± 9.91	11.00 ± 8.46	−0.74	0.460
MMSE	29.77 ± 0.43	29.92 ± 0.27	−1.54	0.135
PA energy expenditure (kcal/day)	66.05 ± 11.97	64.42 ± 10.96	0.50	0.619
Dietary (kcal/day)	2023.1 ± 689.5	1882.26 ± 520.18	0.80	0.429
Circumference				
Waist (cm) *	75.55 ± 3.96	88.68 ± 9.91	−6.26	<0.001
Abdominal (cm) *	84.09 ± 6.12	96.90 ± 10.26	−5.44	<0.001
Hip (cm) *	98.99 ± 4.42	110.64 ± 7.02	−7.12	<0.001
Percentage fat				
Whole body (%) *	34.25 ± 3.89	39.30 ± 4.44	−4.33	<0.001
Upper limbs (%) *	13.53 ± 1.52	14.99 ± 1.66	−3.28	0.002
Trunk (%)	44.44 ± 4.71	46.34 ± 5.60	−1.32	0.194
Lower limb (%)	37.5 ± 5.79	35.46 ± 6.14	1.43	0.159

BMI, body mass index; SBP: systolic blood pressure; DBP: diastolic blood pressure; HR: heart rate; bpm, beat per minute; BDI, Beck depression inventory; MMSE, mini-mental state examination; and PA, physical activity. Values are means ± SD. * *p* < 0.05.

**Table 2 ijerph-17-02726-t002:** Behavioral performance of the normal-weight (control) and obese women.

Characteristics	Control Group (n = 26)	Obese Group (n = 26)
**Go/Nogo task**		
AR Go (%)	99.92 ± 0.23	99.90 ± 0.20
AR Nogo (%)	98.27 ± 2.53	98.40 ± 2.69
RT Go (ms)	479.44 ± 75.42	496.62 ± 72.41
**Stroop task**		
AR congruent (%)	98.88 ± 1.66	99.12 ± 0.93
AR incongruent (%)	97.54 ± 3.08	97.04 ± 2.70
RT congruent (ms)	526.39 ± 64.45	521.20 ± 48.71
RT incongruent (ms)	569.94 ± 97.87	568.62 ± 81.21

AR, accuracy rate; RT, reaction time. Values are means ± SD.

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
