# Peer review of "Neurocognitive Inhibitory Control Ability Performance and Correlations with Biochemical Markers in Obese Women"

_ijerph, 2020, doi:10.3390/ijerph17082726_

Round 1

Reviewer 1 Report

This paper examines the relationship between to inhibitory control measures, markers of inflammation, and changes in ERP components to obesity in women.They found no differences in inhibitory control between healthy and obese women. There were some minor changes in electrophysiology outcomes. However, the main findings are not adequately explored (changes in ERP amplitudes), and without further examination cannot be interpreted with great certainty. The data needs to be explored much further, and the outcome better justified. This could potentially be done with better statistical modelling. My comments below elaborate some minor and major issues with the study.

General

Is it possible that tissue properties, or other factors related to obesity are responsible for the changes in neurocognitive response?

The article is fairly clear and well-written, with definitions of processes and methods well presented.

The study appears well-planned and considered, with clear selection/inclusion criteria and screening out of confounders.

Introduction

*A mention of changes in brain volumes with obesity is an important consideration for interpreting the results. This is not clearly articulated in the discussion.

*Specific mention of the tasks and their relation to anatomical structure and how these change in obesity would be useful

Methods

*Please explain how the participants were recruited

*Please provide more details about the electrophysiology set-up including electrode montage, sampling rate, and equipment models. This study cannot be interpreted without this basic information.

*Is baseline EEG available?

*Please include catalogue numbers for all reagents

Results

*Please report if there were differences in the number of usable averages in the trials

*There are many types of analysis that could be performed on this data – for example including CRP as a covariate in almost all analysis

*Why are no other measures of EEG reported? EEG power would be interesting to examine.

*Attempts to normalise the ERP data should be made – either by peak-peak measurements or ratios of peaks. Currently it is unclear is the differences in size are due to physiological differences, or for example changes in scalp impedance due to increases in subcutaneous fat in obese groups.

*I feel that many of the statistical results would be better presented in table format. Currently it is very hard to read the results due to large number of statistical reporting. I am pleased with this level of detail, but it in this format it is very difficult.

*Furthermore, showing on the Grand Averaged ERPs where the differences lie would be very useful (i.e. with markers of significance “*”, “**”, etc.).

*I am hesitant to base any interpretation on grand-averaged ERPs alone. Please modify to show individual averages, or variability with error bars on all ERP graphs.

*This data needs much more careful consideration. Currently the only current findings (ERP amplitude) do not have a clear neurophysiological meaning. Furthermore, much of this data could potentially be explain by including subject variables as co-variates in their analysis (including BMI, inflammatory biomarkers, head circumference, etc.).

*Also, depending on the electrode density methods such as independent component analysis would be very valuable, and provide better insight into why there are amplitude differences in the ERPs.

*The biomarkers would also benefit from graphical representation.

Discussion

*Please discuss the neurophysiological and anatomical correlates of ERP components. This is very important for interpretation

*Depending on how re-examined results turn out the discussion may need substantial revision. However, currently it is fairly well written.

Reviewer 2 Report

The paper presents a study of the relation between executive function, specifically attention and response inhibition, and obesity. They employ two classic cognitive psychological paradigms, the Go/No-Go task which is evaluates response inhibition and the Stroop task which evaluates attention inhibition. While completing the tasks, electrophysiological measurements were acquired with EEG. These signals were analyzed with emphasis on the N2 and P3 components. The N2 is associated with conflict monitoring and response inhibition. The P3 is a complicated component associated with novelty processing, attention orienting, and response inhibition. It has also been referred to as the “late positive complex”, emphasizing the difficulty in associating the component with a single process. Both components may relate to both tasks, but the there is a large literature relating the N2 and No-Go trials in the Go/No-Go task and the P3 (and later components) with the incongruent trials in the Stroop task.

The obese and control groups were well matched on several factors, including age, height, resting heart rage, depression scores, education, daily physical activity and daily caloric intake. Groups differ on factors related directly to obesity, including weight, body mass index (the defining criteria for obesity in this study), and other physical measurements. The definition of groups was clearly defined, and criteria for sample size selection (power analysis) was reported.

CRP, leptin, and adiponectin levels were assessed via a blood sample and standard assessment kits.

The statistical analyses were conventional and appropriate. However, I feel the paper would have benefitted from reporting standard ANOVA results tables (reporting between and within groups variance, total variance, df, F, p, and partial eta for each factor). If there are concerns about space, I would reduce the size of the ERP figures (perhaps combine as two panels in a single figure).

The authors report neural differences between obese and control groups in the absence of behavioral differences with respect to these tasks. Of these neural differences, only one was related to a biochemical marker: “Only the No-Go P3 amplitude in the obese group was positively correlated with the adiponectin/leptin ratio.”

The authors conclude that there is an association between obesity and inhibitory executive control, and that a possible mechanism is the adiponectin/leptin ratio, by way of insulin. I feel that the conclusions the authors draw are justified, and their arguments in support of their conclusions are well cited and convincing. I do feel like it is worth mentioning that, because there is no behavioral effect, we are stuck with reverse-inference. We know these components are associated with certain processes (e.g. inhibition) and so differences on these components can be taken to imply differences in inhibition. This concern is assuaged, and the overall argument made stronger, by virtue of the tasks chosen being exceedingly well studied and well understood. No one will argue that the Go/No-Go task does not involve inhibitory control. I just point out that even simple tasks are complex, and how exactly these classic tasks relate to ERP components (and vice versa for ERP components and neurocognitive mechanisms). I feel the authors are sufficiently cautious in their discussion.

Of course, the lack of behavioral effects makes one wonder about the real-world impact of these group differences. But the link is certainly plausible, and this relationship is the target of lots of ongoing work. I a strength of the current work is the attempt to relate across behavioral, neural, and biochemical levels.

I want to raise this study to the attention of the authors. It feels highly relevant, and when I came across it, I checked to see if it was cited and it appears not:

Yang, Y., Shields, G. S., Guo, C., & Liu, Y. (2018). Executive function performance in obesity and overweight individuals: A meta-analysis and review. Neuroscience & Biobehavioral Reviews, 84, 225-244.

Otherwise, aside from the request for tables of statistics, I have no further recommendations to improve the manuscript.

Round 2

Reviewer 1 Report

Reviewer 1 Reply to Corrections:

Reviewer 1

Comments and Suggestions for Authors

This paper examines the relationship between to inhibitory control measures, markers of inflammation, and changes in ERP components to obesity in women. They found no differences in inhibitory control between healthy and obese women. There were some minor changes in electrophysiology outcomes. However, the main findings are not adequately explored (changes in ERP amplitudes), and without further examination cannot be interpreted with great certainty. The data needs to be explored much further, and the outcome better justified. This could potentially be done with better statistical modelling. My comments below elaborate some minor and major issues with the study.

General

Is it possible that tissue properties, or other factors related to obesity are responsible for the changes in neurocognitive response?

Reply: Except the fat contents and BMI which were used to divide the participants into the experimental and control groups, previous studies have reported that the cognitive function, depressive status, and blood pressure are related to obesity and responsible for the changes in neurocognitive response. Therefore, we measured the participants’ mental and psychiatric status using Mini-Mental State Examination and Beck depression inventory II and found that there were no significant difference between obese and control groups. However, systolic and diastolic blood pressure achieved significant difference between the two groups (Please see Table 1). In the revised manuscript, the behavioral and ERP performances were separately analyzed again using a repeated-measures analysis of covariance (RM-ANCOVA) to examine the effects of obesity on neurocognitive performance, with the systolic and diastolic blood pressure used as covariates. (Please see the Results section, Lines 319-326, 365-371). Thanks for Reviewer #1’s reminder.

Reviewer Response: Thanks for addressing my comment and including additional factors in your statistical modelling. I was also wondering if you can considered that high levels of adipose tissue on the scalp might change electrical volume-conduction and therefore the amplitude and shape of you ERP signals. This would be worth discussing.

Introduction       

*A mention of changes in brain volumes with obesity is an important consideration for interpreting the results. This is not clearly articulated in the discussion.

Reply: The content, “Indeed, obese individuals demonstrate characteristics of weak inhibitory control, which is considered to play a critical role in difficulties related to resisting external cues suggesting delicious food (Appelhans, 2009; Davis, Levitan, Smith, Tweed, & Curtis, 2006). These behavioral characteristics may be associated with dopamine-modulated mesolimbic circuits and the dorsolateral regions of the prefrontal cortex (Appelhans, 2009). In comparison with normal-weight controls, obese adults have also been demonstrated to exhibit lower dopamine D2 receptor density in the striatum, which is associated with higher metabolic activity in the prefrontal regions involved in inhibitory control and could be a potential mechanism contributing to overeating (Volkow et al., 2008).”, was stated in the introduction section (please see Lines 52-59). We used the brain neural networks related to the obesity to interpret the results in the discussion section (Please see Lines 431-442).

Reviewer Response: Thank you for your clarification. However my my main consideration was of what evidence there is of brain volume and brain signals changes with obesity in neuroimaging such as fMRI, qEEG, MEG, or MRI in obese women, especially in the brain regions and circuits involved in inhibitory control? Changes in neuroimaging observed in the literature would provide insight into why you find changes in ERP amplitude and latency.

*Specific mention of the tasks and their relation to anatomical structure and how these change in obesity would be useful

Reply: The contents regarding the two cognitive tasks and their relation to anatomic structure were stated in the introduction section in the 1st version of the manuscript. (Please see Lines 52-59 & Lines 63-67 as follows)

“Indeed, obese individuals demonstrate characteristics of weak inhibitory control, which is considered to play a critical role in difficulties related to resisting external cues suggesting delicious food (Appelhans, 2009; Davis, Levitan, Smith, Tweed, & Curtis, 2006). These behavioral characteristics may be associated with dopamine-modulated mesolimbic circuits and the dorsolateral regions of the prefrontal cortex (Appelhans, 2009). In comparison with normal-weight controls, obese adults have also been demonstrated to exhibit lower dopamine D2 receptor density in the striatum, which is associated with higher metabolic activity in the prefrontal regions involved in inhibitory control and could be a potential mechanism contributing to overeating (Volkow et al., 2008).” “In the present study, two inhibitory tasks, Go/Nogo and Stroop, were adopted to be as homogeneous as possible in terms of visual-spatial presentation and motor requirements. The two cognitive tasks activate different task-relevant fronto-cingulo-striatal neural networks, i.e., predominantly right fronto-striatal regions during the Go/Nogo task and left-hemispheric parieto-temporal and fronto-striatal regions during the Stroop task (Rubia et al., 2006).”

Reviewer Response: Thanks, this has been adequately addressed. However, additional information about the peaks and their relationship with brain structures would be preferable.

Methods

*Please explain how the participants were recruited.

Reply: The recruitment method has been added in method as the Reviewer #1 suggested. (Please see Lines 126-127)

Reviewer Response: Thank you, this is much appreciated.

*Please provide more details about the electrophysiology set-up including electrode montage, sampling rate, and equipment models. This study cannot be interpreted without this basic information.

Reply: Based on the Reviewer #1’s suggestion, the details about the EEG set-up and equipment models have been added in the revised manuscript. (Please see Lines 204-206).

Reviewer Response: Thank you this is much appreciated, but more details are needed (see author guidelines).

*Is baseline EEG available?

Reply: The baseline EEG (-200 ms-0 ms) was described in the “2.6. ERP recording and analysis” section. (Please see Lines 209-212) “ The remaining effective ERP data were separately averaged offline and constructed from Go and Nogo conditions in the Go/Nogo task and from congruent and incongruent conditions in the Stroop task over a 1000 ms epoch beginning 200 ms prior to the onset of the target stimulus.“

Reviewer Response: Thank you – I take this to mean that no resting-state EEG was recorded. For future studies, a 10 min baseline may be valuable for you and considerably add information with minimal extra effort.

*Please include catalogue numbers for all reagents

Reply: As the Reviewer #1’s suggestion, the catalogue numbers for all reagents have been added. (Please see Lines 222-225)

Reviewer Response: Thank you this is much appreciated, but more information is needed (see author guidelines).

Results

*Please report if there were differences in the number of usable averages in the trials

Reply: Based on the reviewer #1’s suggestion, the number of usable average in the trials are reported below. Obese and control groups did not show significant difference in the mean number of trials that entered into each of the average for RTs (Go/Nogo task: Go trials, p = 0.75 & Nogo trials, p = 0.79; Stroop task: congruent trials, p = 0.47 & incongruent trials, p = 0.50), and for ERP analysis (Go/Nogo task: Go trials, p = 0.54 & Nogo trials, p = 0.65; Stroop task: congruent trials, p = 0.91 & incongruent trials, p = 0.94). 

Reviewer Response: Thank you this is much appreciated. Please incorporate this into the manuscript or supplementary materials. It is incredibly value information. I recommend mentioning this in your results, and then providing the table in supplementary materials.

*There are many types of analysis that could be performed on this data – for example including CRP as a covariate in almost all analysis

Reply: The biochemical marker, CRP, was an important dependent variable which was used to examine the difference between the obese and control groups. Until now, no study demonstrated this molecular biomarker is a confounding factor with regard to the neurocognitive performance of the Go/Nogo and Stroop tasks in individuals with obesity. In addition, one of the main purposes was to explore the correlations between the neurocognitive and inflammatory indices. Therefore, we did not use the CRP (and also other pro-inflammatory cytokines) as a covariate in all analysis.

Reviewer Response: This has been adequately addressed.

*Why are no other measures of EEG reported? EEG power would be interesting to examine.

Reply: The main purposes of the current study was to compare the difference in EEG event-related potentials performance in normal-weight and obese women when performing the Go/Nogo and Stroop tasks, as well as their correlation with biochemical markers. Of course, EEG power (e.g., alpha, beta, theta activities) could be another EEG signal to understand the potential difference between the two groups. To avoid deviating from the main purpose in the present study, the authors did not combine the two EEG signals (i.e., ERP and EEG power) in the same work. We appreciate the Reviewer #1’s suggestion and will consider such an issue in the future study.

Reviewer Response: This has been adequately addressed.

*Attempts to normalise the ERP data should be made – either by peak-peak measurements or ratios of peaks. Currently it is unclear is the differences in size are due to physiological differences, or for example changes in scalp impedance due to increases in subcutaneous fat in obese groups.

Reply: According to Medic et al.’s (2016) study exploring the associations between BMI and cortical thickness in a sample of 203 adults, they found that “There were no associations between BMI and global measures of brain health, such as average cortical thickness or total surface area.” It thus proposed that the subcutaneous fat among the participants would not influence the results of this work. For the reason, we hope that Reviewer #1 will agree with our thinking on this issue.

*Medic N., Ziauddeen H., Ersche K. D., Farooqi I. S., Bullmore E. T., Nathan P. J., et al. . (2016). Increased body mass index is associated with specific regional alterations in brain structure. Int. J. Obes. 40, 1177–1182. 10.1038/ijo.2016.42

Reviewer Response: This has been adequately addressed.

*I feel that many of the statistical results would be better presented in table format. Currently it is very hard to read the results due to large number of statistical reporting. I am pleased with this level of detail, but it in this format it is very difficult.

Reply: The table has been added to report the statistical results as the Reviewer #1 suggested. (Please see appendix)

Reviewer Response: This has been adequately addressed.

*Furthermore, showing on the Grand Averaged ERPs where the differences lie would be very useful (i.e. with markers of significance “*”, “**”, etc.).

Reply: As the Reviewer #1 suggested, the difference markers of amplitudes and latencies have been added in the Grand Averaged ERPs in Figure 1 (between the Lines 296-297) and Figure 2 (between the Lines 347-349). The yellow marker denotes the significant between-group difference in amplitude; and the purple-pink marker in latency. (Please see Figures 1 & 2)

Reviewer Response: This has been adequately addressed.

*I am hesitant to base any interpretation on grand-averaged ERPs alone. Please modify to show individual averages, or variability with error bars on all ERP graphs.

Reply: The averages and standard deviations of ERP N2 and P3 waves in the Go/Nogo and Stroop tasks have been provided below as the Reviewer #1 suggested. The significant difference between obese and control groups have been marked in grand-averaged ERPs in modified Figures 1 and 2.

Reviewer Response: Thanks for the response. This data would actually be best presented in a bar-graph format with error bars and incorporated into the main text in a similar form to Figure 3.

*This data needs much more careful consideration. Currently the only current findings (ERP amplitude) do not have a clear neurophysiological meaning. Furthermore, much of this data could potentially be explain by including subject variables as co-variates in their analysis (including BMI, inflammatory biomarkers, head circumference, etc.).

Reply: As mentioned above, except the fat contents and BMI which were used to divide the participants into the experimental and control groups, the cognitive function, depressive status, and blood pressure are related to obesity and responsible for the changes in neurocognitive response. Therefore, we measured the participants’ mental and psychiatric status using Mini-Mental State Examination and Beck depression inventory II and found that there were no significant difference between obese and control groups. However, systolic and diastolic blood pressure achieved significant difference between the two groups. (Please see Table 1). In the revised manuscript, the behavioral and ERP performances were separately analyzed again using a repeated-measures analysis of covariance (RM-ANCOVA) to examine the effects of obesity on neurocognitive performance, with the systolic and diastolic blood pressure used as covariates. (Please see the Results section, Lines 319-326 and Lines 365-371). In addition, according to previous Medic et al.’s (2016) study exploring the associations between BMI and cortical thickness in a sample of 203 adults, they found that “There were no associations between BMI and global measures of brain health, such as average cortical thickness or total surface area. It thus proposed that the subcutaneous fat among the participants would not influence the results of this work.

*Medic N., Ziauddeen H., Ersche K. D., Farooqi I. S., Bullmore E. T., Nathan P. J., et al. . (2016). Increased body mass index is associated with specific regional alterations in brain structure. Int. J. Obes. 40, 1177–1182. 10.1038/ijo.2016.42

Reviewer Response: This has been adequately addressed.

*Also, depending on the electrode density methods such as independent component analysis would be very valuable, and provide better insight into why there are amplitude differences in the ERPs.

Reply: The main purposes of the present study were to compare the difference in neurocognitive performance in healthy sedentary normal-weight and obese women when performing the Go/Nogo and Stroop tasks, as well as their correlation with biochemical markers. Therefore, we used the same ERP analysis (as adopted in the previous studies) to examine the difference between the two groups. By this way, we can compare the present findings with those in the previous obesity-and-cognitive electrophysiology studies. For the reason, we hope that reviewer #1 will agree with our thinking on this issue.

Reviewer Response: This has been adequately addressed – though much extra value could be obtained from this study by using commonly used toolboxes such as EEGLAB for ERP source localization and peak analysis.

*The biomarkers would also benefit from graphical representation.

Reply: The comparisons of the biomarkers were modified using graphical representation as the Reviewer #1 suggested. (Please see Figure 3, between the Lines 377-387).

Reviewer Response: This has been adequately addressed.

Discussion

*Please discuss the neurophysiological and anatomical correlates of ERP components. This is very important for interpretation

Reply: As mentioned above, we used the brain neural networks related to the obesity to interpret the results in the discussion section. (Please see Page 12, the last paragraph & Page 13, the 2nd paragraph)

Reviewer Response: A more specific reference of different components with their proposed source studies would be valuable. For example with auditory brainstem responses each peak (I to V) has a proposed anatomical source. For your paradigms, I would also expect this is the case and would aid the discussion.

*Depending on how re-examined results turn out the discussion may need substantial revision. However, currently it is fairly well written.

Reply: Most contents in this manuscript has been revised according to the Reviewer #1’s suggestions. Authors hope that reviewer #1 will agree with our responses and manuscript revision.

Reviewer Response: This has been adequately addressed.

Reviewer 2 Report

The N2 and P3 intervals described in the methods section (120–250 ms and 250–400 ms, respectively) do not correspond to the areas of significance in the figures. I failed to notice last time that these stated intervals didn't correspond with the data reported, so I apologize for not catching this on the first round.

In the introduction, the authors state with citations that "N2, an early negative deflection occurring around 200-400 ms post-stimulus, is mainly associated with conflict monitoring processes [19, 20]. The following P3 wave, a positive component occurring around 300-600 ms post-stimulus, is associated with inhibition processing or attentional engagement [21, 22]." This indicates that the ranges indicated in the figure are consistent with prior work you already cite. It seems that the methods section just needs to be revised to reflect the windows that were actually used.

As long as that revision to the methods text is being made, it would be worth mentioning explicitly that ANOVAs are conducted based on mean amplitude within these windows (or, if that is untrue, what ever it is that you actually did--I took it for granted the mean amplitude was used, and didn't notice that it isn't explicitly reported in the document).

Likewise, I took for granted what was meant by N2 or P3 latency, but it would be useful to state something like "Latency was calculated as the time in milliseconds from stimulus onset to peak amplitude."

Given the revisions to the figures done to satisfy reviewer one, the figure captions should be updated to describe what constitutes significance. Since the figures now display statistical inference, the caption should either describe what is being marked as significant or explicitly point me to the text for comprehension.

The amplitude significance highlighting was easy enough to interpret, but I was not sure how to read the latency highlighting. The current solution of highlighting from onset to the peak in one condition doesn't make sense to me (and seems to indicate latency effects in conditions where the figure does not obviously reflect one, based on how I understand "latency").

Personal taste: I don't like the highlighting on the figures. I think it would be cleaner to add vertical dotted lines extending up from the x-axis marking the beginning and end of the N2 and P3 windows. Corresponding time-stamps could be added to the x-axis, making it very clear how windows were defined at a glance. Significance could then be indicated with a conventional *<.05, **<.01, ***<.001. To show latency effects, a horizontal line extending from peak to peak would be sufficient, and this could also be marked with stars for significance.  Just a recommendation. I feel that the highlighting approach looks unpolished. Of course I won't hold up publication based on personal taste, but do feel that the figures could be improved so that the additional information is added in a more intuitive and conventional way.
